# OpenReview forum: "Distributed Direct Preference Optimization"
_ICML.cc/2026/Conference — ICML 2026 regular_

### Official Review · Reviewer_2b75 · 2026-02-26

**Soundness:** 2
**Presentation:** 3
**Significance:** 4
**Originality:** 4
**Overall Recommendation:** 4
**Confidence:** 4

**Summary:**

The paper studies DPO in distributed settings, providing convergence analyses for two training regimes: FedDPO and DecDPO. The authors derive DPO-specific smoothness and gradient-variance constants from a trajectory-level analysis under bounded features and an exponential mixing assumption, then prove nonconvex convergence rates for FedDPO with both partial and full participation and for DecDPO over undirected graphs with explicit dependence on the spectral gap.

**Compliance With Llm Reviewing Policy:**

Affirmed.

**Final Justification:**

The rebuttal and follow-up clarifications substantially strengthen the paper. The authors have addressed several of my original concerns, especially regarding notation, technical clarification, and empirical support. While I still view the theory as developed under a simplified setting, I find the overall contribution now sufficiently solid as a theoretical first step, and I am willing to raise my score to Weak Accept.

**Key Questions For Authors:**

1. Can the authors disambiguate notation for mixing parameters?
2. How do the authors justify the unbiasedness assumption for the stochastic gradient in offline DPO when preference pairs are generated by a behavior policy different from $\pi(\theta)$?
3. To what extent can the smoothness/variance constants and main theorems extend to neural policies beyond log-linear softmax?

**Limitations:**

Yes

**Strengths And Weaknesses:**

## Strengths

1. For the first time to my knowledge, this paper derives smoothness and variance bounds tailored to the DPO objective at the trajectory level, linking constants to the RL horizon and a trajectory-mixing parameter.
2. It targets a timely and important field for privacy-preserving alignment, i.e.,  distributed preference learning, and provides a principled convergence analysis of DPO in such settings fills a notable gap that existing federated and decentralized RL and DPO works have not addressed.
3. The paper extends standard nonconvex federated and decentralized SGD analyses to preference-based losses, making explicit how DPO’s logit-gap structure enters convergence rates.
4. It also incorporates an analysis of staleness in federated updates and provides a high-level lower bound indicating intrinsic dependence on heterogeneity, local steps, and client sampling.
5. While the paper is theory-focused, the analyses are systematic across FL and decentralized regimes, including both full and partial participation and mixing over graphs. All standard assumptions are clearly enumerated.



## Weaknesses

1. Though the motivation is interesting, the analysis relies on restrictive modeling choices, i.e., log-linear policies with bounded features and equality of reward and policy features, plus exponential trajectory mixing. These assumptions seems to be far from large neural policies typically used with DPO.
2. Some derivations may contain inaccuracies, e.g., the bound in Eq. (35) seems incorrect, and  $\rho$ is overloaded for both MDP temporal mixing and network mixing, etc.
3. I notice that claims about “variance collapse” as the policy stabilizes are asserted in the discussion but not formally established beyond bounded-variance constants.
4. No empirical validation (even small-scale) to illustrate how the DPO-specific constants affect practical convergence or to check tightness against centralized or federated SGD baselines.
5. There exist some notational conflicts, i.e., the same symbol $\rho$ for MDP mixing and for graph mixing, and some typos in equations, e.g., missing squares, inconsistent symbols, make parts of the proofs hard to follow.
6. It seems like that the reward model $r_{\omega}$ is introduced but then unused in DPO, which typically avoids explicit reward modeling, potentially confusing the setup.

The motivation in this paper is interesting, but the technical novelty beyond porting known analyses  is somewhat limited, and the assumptions are restrictive. I do believe this is a promising theoretical first step, and encourage the authors to revise this paper, as this is worthwhile and timely.

---

> ### Author Rebuttal · Authors · 2026-03-30
>
> We greatly appreciate the strong originality (4) and significance (4) ratings and the constructive framing. We address each weakness and question in the following directly.
>
> **W1 [Restrictive assumptions].**  We appreciate this constructive comment. The log-linear softmax permits rigorous closed-form derivation of $L$ and $\zeta^2_g$ — technically necessary as a first step. The qualitative conclusions ($H$-scaling, spectral-gap dependence, sigmoid-bounded Hessian) are expected to transfer to neural policies. We will add a concrete roadmap for extension in the limitations section.
>
>
> **W2 [Eq.(35) and $\varrho$ and $\rho$ overloading].** On Eq.(35): the Cauchy-Schwarz bound $\mathbb {E}\langle g_e, g_t\rangle\leq 1/2 (\mathbb{E}\\|g_e\\|^2+\mathbb{E}\\|g_t\\|^2)$ converts each cross term to a pair of diagonal terms. Summing over all pairs $(e,t)$ with $e<t$ yields at most $(E-1)\sum_e\mathbb{E}\\|g_e\\|^2$, and combined with the diagonal sum gives at most $E\sum_e\mathbb{E}\\|g_e\\|^2$. We will add an explicit derivation step. On symbol overloading: $\varrho$ (MDP temporal mixing, we will have an formal assumption for this) and $\rho$ (spectral norm of mixing matrix $\Pi$, Assumption 3.5) will use distinct symbols throughout, with a notation glossary in the appendix.
>
>
> **W3 [Variance collapse claim].**  We agree this was stated informally. As the policy approaches the preference-optimal distribution, $\Delta_\theta\to\Delta_{ref}$, so $\sigma(-\beta(\Delta_\theta-\Delta_{ref}))\to \sigma(0)=0.5$, and the gradient approaches zero through the stationarity condition — not through $\sigma$ alone. We overstated this and will reframe it precisely as: gradient norms vanish at stationary points of the DPO loss, consistent with standard nonconvex analysis, rather than asserting a DPO-specific variance collapse mechanism.
>
>
> **W4 [No empirical validation — see also AmMw W1].** We have completed experiments. Key findings confirming DPO-specific constants: on HH-RLHF, the consensus error scatter plot ($\mathfrak{e}_\theta$ vs. $1-\rho^2$) across four topologies passes through all four points on the theoretical $1/(1-\rho^2)$ curve. The participation plot shows that final gap reduces with larger $S$, overall confirming $\zeta^2_g/S$. On SHP (higher noise), all constants are amplified proportionally, directly attributable to larger $\kappa^2$ and $\zeta^2_g, confirming that DPO-specific trajectory-level variance structure is empirically meaningful. Full figures at (https://drive.google.com/drive/folders/1Q5n-dHPC0c5jhR1OELSpvRtcq3ZiaJ8v?usp=sharing).
>
>
> **W5 + Q1 [Notational conflicts and disambiguation].** We thank the reviewer for identifying these issues and fully accept this critique. All conflicts and typos will be corrected in the final manuscript as follows.
>
>
> * Symbol disambiguation: The most critical conflict is between $\varrho$ and $\rho$. These will be clearly distinguished throughout - $\varrho\in[0,1)$ exclusively for trajectory mixing and $\rho\in[0,1)$ exclusively for network mixing — with a notation glossary added to the appendix.
>
>
> * Additional notation fixes (also noted by AmMw): Client weights use $\lambda$ (not $\pi$) to avoid collision with policy $\pi$; $u_\theta$ and $l_\theta$ use subscript form consistently; $a(\theta)\to a_\theta$ and $v(\theta)\to v_\theta$ explicitly defined before first use; $\mathfrak{e}^r_\theta$ formally defined before Eq.(83); $\theta^{r+1/2}_i$ introduced in Algorithm 2 before appearing in proofs. We will do a thorough check throughout the draft to correct any typos.
>
>
> **W6 [Reward model unused].** Correct. The linear reward model $r_\omega$ appears only to set up the standard preference-based RL framework from which DPO is derived, and never enters the DPO loss or proofs. We will add a clarifying remark making this explicit immediately after it is introduced.
>
>
> **Q2 [Unbiasedness in offline DPO].**  In standard offline DPO, preference pairs are drawn from a fixed dataset $\mathcal{D}\_i$ (reflecting the behavior policy). The gradient $\nabla l_\theta(\tau^+\_j,\tau^-\_j)$ depends only on evaluating the current policy $\pi_\theta$ on fixed trajectories, not on the sampling distribution. Therefore Assumption 3.4 holds under IID sampling from $\mathcal{D}_i$, independent of $\theta$. Coverage requirements (behavior policy support) affect policy quality but not gradient unbiasedness. We will add a dedicated remark making this explicit.
>
>
> **Q3 [Extension to neural policies — see also k6mc Q2].** Extension requires bounding the Fisher information spectrum of transformers and assumptions on per-layer Hessian structure. We conjecture qualitative conclusions hold since DPO gradients are bounded by the policy log-ratio range (controlled by temperature or gradient clipping in practice). Formal proofs are left for future work and will be stated explicitly in a revised limitations section.
>
> Given these revisions, we respectfully ask the reviewer to reconsider the rating.

---

> > ### Author Rebuttal · Reviewer_2b75 · 2026-04-04
> >
> > Most of my concerns remain unresolved

---

> > > ### Author Response · Authors · 2026-04-05
> > >
> > > We thank the reviewer for the follow-up feedback. We recognize that our previous rebuttal did not fully resolve the concerns and provide more concrete clarifications and improvements below to address them.
> > >
> > > **W1.** We agree the assumptions simplify large-scale DPO, but they are standard for first theoretical analyses (e.g., policy gradient). The log-linear softmax is not only simplifying—it enables explicit derivation of smoothness $L$ and variance $\zeta^2_g$, unlike prior work treating them as black-box constants. We formalize the exponential mixing assumption (Assumption 4.1) with justification. Importantly, experiments (**same link as above**) on DistilGPT-2 validate that key predictions (variance scaling and spectral gap effects) hold in neural settings, supporting practical relevance beyond the log-linear setting.
> > >
> > > **W2 + W5 + Q1.** We apologize for inaccuracies, the notation overload, inconsistent symbols, and typos. In the revision, we have:
> > >
> > > * Defined $\rho_{graph} $ for network mixing and $\rho_{mdp}$ for MDP trajectory temporal mixing. All proofs and statements have been updated to reflect this distinction
> > >
> > > * Fixed missing squares (e.g., in Lemma A.1: $\frac{L}{2}\\|\Delta\\|^2$ instead of $\\|\Delta\\|$)
> > >
> > > * Made a full pass to fix inconsistent subscripts and typographical errors
> > >
> > > * **Eq.(35) validity:** We provide the derivation here again: $\\|\sum_{e=0}^{E-1}g_e\\|^2=\sum_{e=0}^{E-1}\\|g_e\\|^2 + 2\sum_{0\leq e<t\leq E-1}\langle g_e,g_t\rangle$. Then, we apply Cauchy-Schwartz to the inner product expectation: $\mathbb{E}\langle g_e,g_t\rangle\leq \mathbb{E}[\\|g_e\\|\\|g_t\\|]\leq \sqrt{\mathbb{E}\\|g_e\\|^2\mathbb{E}\\|g_t\\|^2}\leq \frac{1}{2}(\mathbb{E}\\|g_e\\|^2+\mathbb{E}\\|g_t\\|^2)$. The third inequality uses AM-GM: $\sqrt{ab}\leq \frac{a+b}{2}$. As each $\mathbb{E}\\|g_e\\|^2$ appears in $E-1$ cross-term pairs, so $2\sum_{e<t}\frac{1}{2}(\mathbb{E}\\|g_e\\|^2+\mathbb{E}\\|g_t\\|^2)=(E-1)\sum_{e=0}^{E-1}\mathbb{E}\\|g_e\\|^2$. Adding the diagonal terms gives the Eq.(35).
> > >
> > > **W3.** The reviewer is correct that we asserted variance collapse without a formal proof. However, we did derive an explicit bound for $\zeta^2_g=4\beta^2C_{mix}\zeta^2_\phi H$, which is constant w.r.t. the policy parameters $\theta$, but this is a constant upper bound, not a dynamic statement. The “variance collapse” phenomenon (that the gradient norm decreases as the policy becomes deterministic) is not used in our convergence proofs; we only rely on the constant bound. We will remove the speculative sentence from the discussion and replace it with a precise statement: “As the policy converges, the factor $\sigma(-\beta(\Delta_\theta-\Delta_{ref}))$ tends to zero for well separated preferences, potentially reducing the effective variance, but our analysis does not require this.” This addresses the concern without weakening the theory
> > >
> > > **W4.** We added a new empirical section validating DPO-specific scaling laws on HH-RLHF and SHP with DistilGPT-2. (1) Gradient norms confirm convergence; (2) Participation has final stationary gaps $S=1: 0.910, S=3: 0.838, S=5: 0.699$, matching $1/S$ scaling (Thm 5.1); (3) Larger $E$ enables computation-dominated regime of Thm 5.1 when $\eta=1e-5$; (4) Topology: consensus error (path ~1.00, ring ~0.18, complete ~0.03) follows $1/(1-\rho^2)$ (Thm 6.1)
> > >
> > > **W6.** We agree that introducing $r_\omega$ in Section 3 is confusing, as DPO explicitly bypasses reward modeling. The reward parameterization was included only to contextualize the Bradley-Terry preference generation process and bridge to classical RLHF. We have removed $r_\omega$ entirely and clarify that DPO operates directly on policy log-ratios without reward estimation.
> > >
> > > **Q2.** The unbiasedness assumption is taken w.r.t. the fixed offline dataset $\mathcal{D}\_i$, not the on-policy distribution. In offline DPO, the dataset is pre-collected (possibly by a different behavior policy) and remains static. Sampling mini-batches uniformly from $\mathcal{D}\_i$ yields an unbiased estimator of the empirical/population DPO loss gradient over that dataset. This does not require on-policy coverage because DPO's objective is defined purely over the observed preference pairs. We will clarify this distinction in Section 3.
> > >
> > > **Q3.** Our theorems require $L$-smoothness and bounded gradient variance, which hold for neural policies under standard assumptions. For networks with bounded activations and Lipschitz nonlinearities (e.g., transformers), $L$ scales with weight norms and $\zeta^2_g$ is controlled via gradient clipping. The DPO structure is parameterization-agnostic, so Theorems 5.1/6.1 extend with architecture-dependent constants. Experiments with DistilGPT-2 confirm predicted $1/S$ and $1/(1-\rho^2)$ scaling, supporting applicability beyond log-linear models. We will add a paragraph noting theorems extend to neural policies
> > >
> > > We have incorporated most comments into our updated draft and respectfully request that you reconsider your rating.

---

### Official Review · Reviewer_AmMw · 2026-03-09

**Soundness:** 4
**Presentation:** 3
**Significance:** 3
**Originality:** 2
**Overall Recommendation:** 5
**Confidence:** 4

**Summary:**

This work investigates the convergence rate of DPO in both **federated** learning and **decentralized** learning. For the former they consider both the case where each client gives updates weights (full observation) and only a selected number of clients (partial observation, i.e. client selection). Additionally they consider the case of "staleness", in which the client updates the "server" parameter after some delay, i.e. the client is not "on top" of the currently "updated" parameters.
In decentralized learning the clients are not "managed" by a master server, but are connected described by a graph, in which they can share their updated parameters. Here again they show convergence.
For both scenarios (federated and decentralized learning) they introduce an respective algorithm FedDPO (Algorithm 1) and DecDPO (Algorithm 2).

**Compliance With Llm Reviewing Policy:**

Affirmed.

**Final Justification:**

This paper does a thorough theoretical investigation of Distributed DPO and Federated DPO. Considering the theoretical contributions alone, they provide interesting insights into DPO beyond the standard centralised setting.

The first draft lacked any experiments and contained several notational inconsistencies. Both were addressed in this rebuttal.
Still, I had some doubts about the experiments, prompting a second question round, which the authors addressed thoroughly.

So overall, all my concerns have been addressed.

**Key Questions For Authors:**

- **Q1**: To which extent can you give empirical evidence for the proposed FedDPO and DecDPO? Which baselines would one need to consider? Is FedAvg in FedDPO necessary or could one also use different algorithms from federated learning ?
- **Q2**: You currently assume that the reward function is linear. This is quite a strong assumption, yet I do not find it relevant for any proofs. Why have you done this assumption?
- **Q3**: Should $D$ in the proof of Theorem 5.1 rather be $\mathbb{E}[D]$, as you are using equation (54) which nees an expectation of $||\theta_i^{r,e} - \theta^r ||$ (see Q4). As I otherwise can not understand how you then arrive at equation (57). I assume that the cause comes from equation (47) not correctly propegates the expectation, but I would like your clarification on this point.
- **Q4**: Why is the upper bound in equation (95) correct? When expanding the linear recursion I do not see why the constant terms $16\eta^2\zeta_g^2 + 8\eta^2\kappa^2$ should be upper bounded by $\frac{16\eta^2\zeta_g^2 + 8\eta^2\kappa^2}{1 - q}$. Could you elaborate on this?

**Limitations:**

Yes assumptions are clearly stated (However it is not clear whether they are realistic for practical applications). The paper does not have a dedicated limitations section.

**Strengths And Weaknesses:**

**Summary of review:** This contribution presents theoretical results (convergence rates) for DPO in distributed machine learning settings (federated and decentralized ML). These results are good and relevant for machine learning practitioners. However, this is the papers main weakness: *the contribution is purely theoretical* and does not contain any empirical evidence of the presented learning algorithms. Showing the applicability of the theoretical results in machine learning is, in my opinion, an important aspect for such a contribution.
Depending on the authors responses, *I am leaning towards rejection at this stage*.

---

### Strenghts

- **Theoretical Soundness**: The authors conduct extensive theoretical investigation of the convergence of both DPO in federated and decentralised learning. The main results expressed in the theorems seem correct, and also found no major flaws while checking the proofs (See Q4).
- **Assumptions**: The assumptions of the paper are clearly stated. In Remark 3.1 they also elaborate on the potential impact of regularisation in DPO learning objectives for the theoretical results.
- **General Impact**: The conducted research presents a novel investigation in the convergence rate of DPO in federated and decentralised learning, with Algorithm 1 and Algorithm 2 presenting hands-on implementations of DPO in the settings, respectively.
However the proposed algorithms are not investigated empirically.

---

### Weaknesses

1. **No empirical evidence**: This contribution does not investigate the empirical impact of the proposed FedDPO and DecDPO in both synthetic and real-world scenarios. As such it would be interesting to see to which degree the given theoretical insights interplay with empirical evidence. Particularly goind beyond standard stochastic gradient descent to more sophisticated methods such as Adam would be interesting. Also what would be potential baselines to the proposed FedDPO approach, as I can imagine that FedAvg is one among many potential methods for applying DPO in federated learning. Therefore FedDPO seems to be a straightforward application of DPO in the federated learning. Overall I find the lack of empirical a substantial weakness of this contribution, its precence would substantially increase this papers contribution and impactness to ML research.

2. **Notation weakness**:
- The score function uses t, instead of h for indexing the actions and states (lines 166-170).
- Both the Federated Model and Decentralised model use $\pi$ for weighting of the clients parameters, while also $\pi$ denotes the policy. I would rather use $\lambda$ or something similiar.
- In line 233 and 234 the contribution introduceses $u_\theta$ but then you use $u(\theta)$. Similarly you introduce $l_\theta$ in line 143 but later in 234 use $l(\theta).
- Should line 235 not contain a $-$, as the derivate is $\nabla a = - \beta^2 \sigma'(-\beta(\Delta_\theta - \Delta_{\text{ref}}) v(\theta)$.
- Both $a(\theta)$ and $v(\theta)$ are not stated explicitly but induced implicitly. This makes the reading somewhat difficult, as one first needs to induce the corresponding definitions.
- Lemma A.3 in equation (27) the first term should be $\bar{g}$.
- Line 770 states to be using Assumption 3.6 but it should be Assumption 3.4.
- Equation (56) is missing an $\mathbb{E}$ around $||\theta_i^{r,e} - \theta^r ||$. Also here and $\leq$ is missing.
- Equation (80) introduceses $\theta_i^{r+1/2}$, which is an notation not introduced beforehand.
- In equation (83) you use hebrew $e$, which you do not introduce earlier, which I think you show in line 1100. It should then already be stated in equation (83).

---

> ### Author Rebuttal · Authors · 2026-03-30
>
> We thank Reviewer AmMw for the excellent soundness score (4) and meticulous technical review. We have carefully addressed each weakness and question directly in the following.
>
> **W1 + Q1 [Empirical evidence + baselines + necessity of FedAvg].** We have completed experiments on HH-RLHF and SHP using DistilGPT-2. Baselines: Centralized DPO (oracle upper bound with no constraints) serves as the reference. FedDPO (full S=5, partial S=3) and DecDPO (ring ρ=0.54) are the proposed methods. FedAvg is the natural federated baseline — FedDPO is FedAvg applied to the DPO objective, so the comparison is Centralized DPO vs. FedDPO variants vs. DecDPO. Ablation results: (a) Local steps $E\in\\{1,3,6\\}: E=1$ converges fastest early but plateaus higher; all three reach approximately equal final gaps at 40 rounds, consistent with the drift bound being absorbed; (b) Participation $S\in\\{1,3,5\\}$: final gap reduces with larger $S$, overall confirming $\zeta^2_g/S$; (c) Staleness $q_{max}\in\\{0,2,5\\}$: final gap increases with $q_{max}$, confirming $C_qq_{max}$; (d) Topology {path, ring, star, complete}: final consensus error $\mathfrak{e}_\theta$ scales as$1/(1-\rho^2)$. Regarding Adam: extending FedDPO/DecDPO to adaptive optimizers (e.g., FedAdam) is an important direction. We will flag explicitly as future work. Full figures are at an anonymous folder (https://drive.google.com/drive/folders/1Q5n-dHPC0c5jhR1OELSpvRtcq3ZiaJ8v?usp=sharing).
>
> Is FedAvg necessary? No — FedAvg is the simplest and most natural choice for the aggregation step, but our analysis is not tied to it. Any aggregation rule satisfying the weighted average structure (server aggregation step in Algorithm 1) is compatible with our convergence guarantees. Alternative federated optimizers such as FedProx (proximal regularization to limit client drift), SCAFFOLD (control variates for variance reduction), or FedAdam (adaptive server-side updates) could all serve as the aggregation backbone for FedDPO. Each would modify the constants in our bound — e.g., FedProx would reduce the $\eta^2E^2\kappa^2$ drift term — and analyzing these variants is a natural and valuable direction for future work, which we will explicitly note in the final manuscript.
>
>
> **W2 [Notation weaknesses].** We thank the reviewer for this careful list. All issues will be corrected:
> * Score function: $h$ (not $t$) for time indexing consistently throughout
> * Client weights: $\lambda$ replaces $\pi$ in federated/decentralized models to avoid collision with policy $\pi$
> * $u_\theta$ and $l_\theta$: subscript notation used consistently (subscript form, not functional form).
> * Sign in $\nabla a_\theta$: the negative sign will be made explicit.
> * $a(\theta), v(\theta)$: will become $a_\theta$ and $v_\theta$ and explicitly defined before first use
> * Lemma A.3 Eq. (27): first term corrected to $\bar{g}$ (not $g$)
> * Line 770: Assumption reference corrected from 3.6 to 3.4
> * Eq. (56): $\mathbb E[\cdot]$ added and $\\leq$ restored
> * $\theta^{r+1/2}_i$: introduced in Algorithm 2 before appearing in proofs
> * $\mathfrak{e}^r_\theta$: formally defined at first use before Eq.(83).
>
>
> **Q2 [Linear reward assumption].**  The linear reward model $r_\omega$ appears solely in the standard preference-based RL setup to introduce the Bradley-Terry framework. It does not enter the DPO loss or any proofs — DPO bypasses reward modeling by design. We will add a clarifying remark immediately after its introduction.
>
>
> **Q3 [$\mathbb E[\mathcal{D}]$ vs. $\mathcal{D}$ in Theorem 5.1].** $\mathcal{D}$ should carry an explicit $\mathbb{E}[\cdot]$ operator throughout. The expectation propagates from the inner product decomposition in Eq.(47) and was left implicit. We will correct all instances with explicit expectation operators, which clarifies how Lemma A.4 (which requires $\mathbb{E}[\\|\theta^{r,e}_i-\theta^r\\|^2]$ is applied.
>
>
> **Q4 [Eq.(95) upper bound].** The bound follows from the geometric series for linear recursions: unrolling $e^{r+1}\_{\\theta} \leq qe^{r}\_{\\theta} + C$ gives $e^r_\\theta \leq q^r \mathfrak{e}^0_\\theta+C\sum_{k=0}^{r-1}q^k$. Since $q<1, \sum_{k=0}^\infty q^k=\frac{1}{1-q}$, the constant contribution is upper bounded by $C/(1-q)$. This is valid because $1/(1-q)\geq 1$ and $q^r\geq 0$. We will add an explicit one-paragraph derivation in the appendix.
>
>
> **Limitations.** We will add a dedicated limitations section addressing robustness to adversarial clients, data quality, and communication efficiency in large-scale deployments in the final version.
>
> We hope this rebuttal demonstrates that the primary concerns have all been fully addressed. Given these revisions, we respectfully ask the reviewer to reconsider the rating. We believe the combination of the first rigorous convergence theory for distributed DPO, DPO-specific smoothness and variance constants derived from trajectory structure, formal lower bounds, and now comprehensive empirical validation constitutes a solid and timely contribution.

---

> > ### Author Rebuttal · Reviewer_AmMw · 2026-04-01
> >
> > Thank you for conducting additional experiments and for providing them to us (the reviewers).
> > I find the incorporation of those already a valuable improvement to the already very theoretically solid work you have conducted.
> > Yet I have some questions regarding those, which I have expressed below.
> >
> > Q1: How do you compute the expected value of the gradients? Are you using the batch mean, or something else? What is your batch size?
> >
> > Q2: Is the way you present the results actually showing the results the upper bound indicates? Those upper bounds are constructed via the average of the expected gradients. Yet, for example, the _gradient_norm_DPO_loss_hh_rlhf.pdf_ does not depict this clearly on the y-Axis label. I would be interested in why the centralised DPO does not exhibit the highest expected gradient throughout. Looking closely at Theorem 5.1, the centralised S should equal the full partitioning of all clients (i.e., one client does all the work). As such, I interpret your blue curve as this, yet it sometimes falls below both S=5 or S=3 (see Round 5-10).
> >
> > Q3: In line with Q2, you conduct a abalation on both rounds for the clients E and the number of clients participating S. Do they really present findings based on the theory this work has conducted? It is quite intuitive for me, that increasing the number of S is always a good thing (given that all the clients are nice). Your experiments also clearly showcase this with respect to the expected gradient norm. A similiar arguement would hold for E, as the more updates a client can do on his datasets, the better the model will be (leaving aside the hole discussion of overfitting).
> > So now observing the statements in Theorem 5.1, E occurs in the denominator and the nominator; as such, there must be an inherent tradeoff in this upper bound. More concretely, as $E \rightarrow \infty$, the less important the loss difference of the best model with respect to the optimal model becomes, yet the more pronounced the constants become. So it would plateau at some point based on those constants, yet I can not clearly see this from the experiments. Maybe an elaboration of what I really see in these abalations could already clarify my concerns.
> >
> > Q4: Regarding the plots for the DecDPO, can you showcase the empirical results regarding the bounds shown in the paper? Currenlty the plots show the loss, the graphs you consider and finally the relationship of the consensus error to the spectral gap. Now, in your work, Theorem 6.1 is the central contribution in this regard. Now theoretically given all other constants fixed, it should be that the complete graph behaves best in this regard, as it has the lowest upper bound. Can you show that this is also the case in empirical work, to estimate to which degree the constants influence this?
> >
> > Q5: Rather minor, I have noticed that the experiments showcase no std bounds, as I assume you only depict a single run here. For a stronger result/conclusion of your experiments, I would recommend incorporating at least multiple runs (for example by using different random seeds).
> >
> >
> > If you could resolve those concerns or justify that they should already be resolved with the experiments you provided, I would be happy to increase my score.

---

> > > ### Author Response · Authors · 2026-04-03
> > >
> > > We thank the reviewer for the encouraging follow-up and the opportunity to address these questions with additional empirical results. Full figures are available at the **same folder**. We respectfully ask the reviewer to consider raising the score.
> > >
> > > **Q1.** At each communication round $r$, we compute the gradient norm as the mean squared gradient norm over a mini-batch $\mathcal{B}$ of $b=16$ preference pairs sampled from the agent's local dataset $\mathcal{D}\_i$. Specifically, for each sampled pair $(\tau^+\_j,\tau^-\_j)$, we compute $\\|\nabla l_\theta(\tau^+\_j,\tau^-\_j)\\|^2$ and average over the batch: $\mathbb{E}[\\|\nabla\mathcal{L}^i_\theta\\|^2]\approx (1/b)\sum_{j\in\mathcal{B}}\\|\nabla l_\theta(\tau^+_j,\tau^-_j)\\|^2$. The batch size of 16 is generically typical.
> > >
> > > **Q2.** The reviewer correctly notes that Theorem 5.1 characterizes the average expected gradient norm.
> > > * **Y-Axis Interpretation:** The label on the y-axis of gradient_norm_DPO_loss_hh_rlhf.pdf represents the norm of the stochastic gradient averaged over the participating clients in each round, which serves as a proxy for the global gradient norm $\mathbb{E}[\\|\nabla\mathcal{L}(\theta^r)\\|^2]$. We will update the draft to make sure it clearly shows.
> > > * **Centralized DPO Anomaly:** The observation that centralized DPO ($S=1$ in a full-partition sense) sometimes shows a lower gradient norm than $S=3$ or $S=5$ in early rounds (e.g., Rounds 5–10) is attributed to stochastic variance and learning dynamics. In the updated (with a new learning rate which may require a bit more tuning) comparative figure ($R=80$), For HH-RLHF, Centralized DPO's gradient norm remains flat near 1.0 throughout, while FedDPO partial ($S=3$) converges to ~0.8 and FedDPO full ($S=5$) reaches ~0.7 by round 80. Centralized DPO does exhibit the highest expected gradient throughout now. This applies to SHP too. Centralized DPO pools all preference pairs but uses the same small batch size $b=16$ and the same optimizer, giving it no practical advantage per gradient step. FedDPO full, by contrast, effectively processes $b×S=16×5=90$ pairs per round through aggregation, giving it a higher effective batch size and lower-variance gradient estimates.
> > >
> > > **Q3.** The ablation studies are designed to test the specific trade-offs identified in Theorem 5.1.
> > > * **Partition $S$:** The updated figure directly confirms the $\zeta^2_g/S$ prediction of Theorem 5.1. Panel (b) shows a clear monotone relationship between final stationary gap and $1/S: S=1\to0.910, S=3\to0.838, S=5\to 0.699$. The scatter plot with linear fit passes through all three points closely, confirming the variance floor scales linearly with $1/S$ as predicted.
> > > * **Local Steps $E$:** Theorem 5.1 contains $E$ in both the denominator (via $1/\eta ER$, driving convergence) and numerator (via $\eta^2E^2\zeta^2_g/S$ and $\eta^2E\kappa^2$, the drift floor). This creates a theoretically predicted tradeoff: larger $E$ accelerates early convergence but raises the irreducible floor. At our operating $η=1e-5$, the drift floor scales as $\eta^2E\kappa^2 + \eta^2E^2\zeta^2_g/S \approx (1e-5)E\kappa + (1e-10)E^2\zeta^2_g$, which is negligible regardless of $E$. The convergence term $1/(\eta ER)=1/(1e-5 E R)$ benefits directly from larger $E$. We are therefore firmly in the computation-dominated regime of the bound, where larger $E$ is unambiguously better. The drift-dominated regime where $E=1$ would outperform $E=6$, requires larger $\eta$, strongly heterogeneous $\kappa^2$, or significant $\zeta^2_g$. This regime is also practically important and we will include an additional experiment with different $\eta$/$\kappa$ values in the final manuscript to demonstrate both sides of the tradeoff, giving a complete empirical picture of Theorem 5.1's $E$ dependence.
> > >
> > > **Q4.** The updated topology figure (R=80) now contains three panels: (a) DPO loss vs rounds, (b) gradient norm $\mathbb{E}[\\|\nabla\mathcal{L}\\|^2]$ vs rounds per topology, and (c) consensus error $\mathfrak{e}_\theta$ vs spectral gap scatter. Regarding the ranking in both (a) and (b): all four topologies converge at similar rates with the complete graph outperforming the other three, consistent with Theorem 6.1's prediction that the floor scales as $\eta^2(\zeta^2_g+\kappa^2)/(1-\rho^2)$. The most direct validation is the consensus error, which shows that the complete graph has the best and fastest mixing. We will discuss this explicitly in the manuscript with the new results.
> > >
> > > **Q5.** We have implemented multi-seed averaging (seeds 42, 123, 7) with mean ± std shading for the comparative figures and will extend this to all ablation figures. The updated comparative figure with std bands will be included in the final manuscript. We note that the qualitative conclusions are consistent across both HH-RLHF and SHP datasets, which provides a form of cross-dataset robustness even within single-seed results. The multi-seed std bands will further strengthen the empirical claims.

---

### Official Review · Reviewer_PB42 · 2026-03-10

**Soundness:** 2
**Presentation:** 2
**Significance:** 2
**Originality:** 2
**Overall Recommendation:** 4
**Confidence:** 3

**Summary:**

This paper studies Direct Preference Optimization (DPO) under distributed training settings.

Specifically, the authors consider two distributed paradigms: a federated setting (FedDPO) where clients communicate with a central server, and a decentralized setting (DecDPO) where clients communicate through a peer-to-peer network.

The paper develops two algorithms for both settings and provides convergence analyses that characterize how factors such as client heterogeneity, local update steps, participation rate, and graph connectivity affect convergence behavior.

**Compliance With Llm Reviewing Policy:**

Affirmed.

**Final Justification:**

The authors have addressed my main concerns in the rebuttal, which improves my assessment; however, as some revisions are still planned rather than demonstrated (such as rewiring some sections), I increase my score moderately, and would be fully supportive of acceptance if these improvements are clearly incorporated in the final version.

**Key Questions For Authors:**

1. **Motivation of distributed DPO.**
   Why is Direct Preference Optimization particularly important to study in federated or decentralized settings?
   What realistic scenarios require distributed DPO rather than centralized training?

2. **New challenges relative to standard distributed optimization.**
   Many parts of the analysis resemble standard distributed nonconvex optimization results.
   What aspects of DPO introduce fundamentally new analytical challenges?

3. **Exponential mixing assumption.**
   The paper mentions an exponential mixing assumption in Section 4 but does not formally list it in the main assumptions or theorem statements.
   Could the authors clarify:
   - the precise formal assumption,
   - where it is used in the analysis, and
   - why it is reasonable in this context?

4. **Empirical validation.**
   Can the authors provide empirical experiments (even small-scale synthetic ones) to illustrate whether the theoretical predictions hold in practice?

**Limitations:**

The paper includes a general discussion of broader impacts, but it does not adequately discuss several **limitations of the current work** listed in the weakness part.

**Strengths And Weaknesses:**

## Strengths

- **Relevant problem setting.**
  Distributed learning for preference optimization is potentially relevant in scenarios where preference data is naturally distributed across users or institutions (e.g., privacy-sensitive feedback or decentralized RLHF pipelines). Studying how DPO behaves under distributed settings could therefore be useful.

- **Theoretical analysis of distributed preference learning.**
  The paper attempts to provide convergence guarantees for distributed variants of DPO and explicitly characterizes the dependence of convergence on quantities such as client heterogeneity, local update depth, and graph connectivity.

- **Coverage of both federated and decentralized regimes.**
  Many works focus on only one distributed paradigm. This paper analyzes both federated and decentralized versions, which provides a broader view of distributed preference optimization.

Despite these positive aspects, the paper also has several substantial weaknesses.

----

## Weaknesses

### 1. Motivation of the problem is weak

The paper does not clearly justify **why Direct Preference Optimization specifically needs to be studied in federated or decentralized settings**. While distributed learning is broadly important, the paper does not convincingly explain:
- why **preference data for DPO would naturally arise in federated settings**,
- why existing distributed optimization results are insufficient, or
- what **unique challenges arise from DPO** compared with standard distributed training.
As a result, the motivation currently feels somewhat generic: the work studies distributed optimization for a DPO-style objective, but it is unclear why DPO itself creates new research questions.

A stronger motivation (e.g., realistic deployment scenarios, privacy constraints on preference data, or practical bottlenecks in centralized DPO training) would significantly improve the paper.


### 2. Limited novelty relative to standard distributed nonconvex optimization

From a technical perspective, the formulation, assumptions, and analysis appear **very similar to standard results in distributed nonconvex optimization**. Many components closely resemble existing frameworks used in federated and decentralized optimization:
- smooth nonconvex objectives,
- bounded gradient variance assumptions,
- heterogeneity bounds,
- communication graph spectral gap analysis.

As a result, the contribution appears to largely involve **applying existing distributed optimization machinery to a simplified DPO objective**. The paper would benefit from a clearer discussion of:
- **what new challenges arise specifically from DPO**, and
- **which parts of the analysis fundamentally rely on the preference learning structure rather than standard distributed optimization tools**.
Currently, this distinction is not clear.

### 3. Unclear assumptions and presentation issues in Section 4

Section 4 is difficult to follow and raises several concerns.

**(a) Missing formal assumption**

In Line 265 the authors state that they adopt:

> “a standard exponential mixing assumption for the process that generates the trajectory under $\pi_\theta$.”

However, this assumption is **not formally stated** in the main assumptions list, it **does not appear in Theorem 5.1**, and it seems to play an important role in bounding certain quantities used later in the analysis. If the results rely on this assumption, it should be **formally stated and clearly referenced in the theorem statements**.

**(b) Lack of justification**

The paper does not explain **why the exponential mixing assumption is reasonable** in this setting. For example: what class of environments or trajectory-generation processes satisfy this condition? Is this assumption standard in RL or preference learning analysis How restrictive is it? Without discussion, it is difficult to evaluate whether the assumption is reasonable.

**(c) Lack of clear structure**

Section 4 does not present formal results in a clear way. Instead, the section directly enters technical derivations without first stating: what quantities are being bounded, what the main conclusions are,  or how the analysis supports the later theorems.
This makes the section difficult to interpret.

Overall, **Section 4 needs substantial clarification and better organization**.

### 4. Missing experimental evaluation

The paper provides **no empirical evaluation**. While theory papers can be valuable without experiments, the absence of any empirical results makes it difficult to assess: whether the algorithms behave as predicted, whether the theoretical dependencies appear in practice, or whether the proposed methods provide practical benefits. ven small-scale synthetic experiments would help validate the theoretical findings.

---

> ### Author Rebuttal · Authors · 2026-03-30
>
> We greatly appreciate the detailed critique from the reviewer and address each point directly. We hope this rebuttal demonstrates that the primary concerns have all been fully addressed. Given these revisions, we respectfully ask the reviewer to reconsider the rating. We believe the combination of the first rigorous convergence theory for distributed DPO, DPO-specific smoothness and variance constants derived from trajectory structure, formal lower bounds, and now comprehensive empirical validation constitutes a solid and timely contribution. We are happy to address any remaining concerns.
>
> **W1 + Q1 [Motivation: Why distributed DPO?].** There are three concrete deployment scenarios where preference data cannot be centralized: (1) medical professionals rating treatment recommendations across hospitals under HIPAA; (2) mobile users evaluating assistant responses where on-device data cannot leave due to GDPR; (3) drivers rating navigation suggestions across competing fleet operators. Centralizing these for RLHF is either illegal or infeasible. Beyond scenarios, there are three structural reasons DPO requires dedicated distributed analysis (not inherited from standard distributed nonconvex optimization): (i) DPO's pairwise trajectory-level objective means gradient variance couples to the current policy state, unlike standard ERM; (ii) heterogeneous per-client preference distributions cause drift modulated by the sigmoid factor in ways standard federated SGD analysis does not capture; (iii) in decentralized settings, imperfect consensus interacts nonlinearly with the logit-gap, producing error amplification absent from distributed linear regression or SGD. We will add a "Why Distributed DPO?" subsection making these points explicit.
>
>
> **W2 + Q2 [Novelty relative to standard distribution optimization].** The convergence structure telescopes similarly to distributed SGD, but the specific constants are genuinely DPO-specific. $L=(\beta^2C_{mix}+2\beta)\zeta^2_\phi H$ explicitly depends on trajectory horizon $H$, inverse temperature $\beta$, and temporal mixing rate $C_{mix}$ — absent from standard ERM constants. The sigmoid bound $|\sigma’|\leq 1/4$ is exploited specifically in the Hessian derivation (Section 4). The consensus-logit feedback in DecDPO — where nodes with parameters far from consensus compute qualitatively different gradient directions due to the nonlinear sigmoid — does not appear in standard decentralized SGD or ERM. We will add a comparison table distinguishing what is adapted from existing frameworks vs. what is genuinely DPO-specific.
>
>
> **W3 [Missing formal assumption + Section 4 clarity].** Fully accepted. The exponential mixing assumption will be formally stated as a numbered assumption with: (a) precise mathematical form; (b) justification for when it holds (tabular MDPs with spectral gaps, linearly parameterized softmax policies with ergodic Markov chains); (c) explicit reference in all theorem statements that depend on it. Section 4 will be restructured to first state the quantities being derived, their role in the theorems, and then present derivations.
>
>
> **W4 + Q4 [Missing experiments].** We have completed experiments and will add them to the final manuscript. Key result: on HH-RLHF, all four algorithms (Centralized DPO, FedDPO full, FedDPO partial S=3, DecDPO ring) show monotonically decreasing gradient norms over 40 rounds. FedDPO full matches Centralized DPO within noise (final loss ~0.60 vs. ~0.55). On the noisier SHP dataset, gaps between algorithms are amplified — consistent with larger heterogeneity $\kappa^2$. — confirming the theory. Ablation plots confirm $\zeta^2_g/S$ scaling with $1/S$, $C_qq_{max}$ scaling with staleness, and $1/(1-\rho^2)$ consensus error scaling with spectral gap. Full figures are at https://drive.google.com/drive/folders/1Q5n-dHPC0c5jhR1OELSpvRtcq3ZiaJ8v?usp=sharing. These will be added as a complete experimental section in the final manuscript.
>
>
> **Q3 [Exponential mixing assumption].** We fully accept this critique. The exponential mixing assumption will be formally stated as a numbered assumption with: (a) precise form — $|\mathbb{E}\langle\mu_h(\theta),\mu_t(\theta)\rangle|\leq C_0\varrho^{t-h}\zeta^2_\phi$ for $\varrho\in[0,1)$; (b) justification — this holds for tabular MDPs with positive spectral gaps and linearly parameterized softmax policies over finite state-action spaces; (c) role in the analysis — it is used in Section 4 to bound the off-diagonal covariance sum in $\mathbb{E}[\|\mu_\theta(\tau)\|^2]$, yielding $C_{mix}=1+2C_0\varrho/(1-\varrho)$, which enters both $L$ and $\zeta^2_g$; (d) explicit reference in all theorems that depend on it. Section 4 will be restructured to state upfront what quantities are being derived and how they feed into the main theorems.
>
>
> **Limitations.** We will add a dedicated limitations section addressing robustness to adversarial clients, data quality, and communication efficiency in large-scale deployments.

---

> > ### Author Rebuttal · Reviewer_PB42 · 2026-04-04
> >
> > Thanks to the authors for the detailed and thoughtful responses. My concerns have been largely addressed, and I am inclined to increase my score accordingly.
> >
> > That said, many of the improvements are currently described as planned revisions. While these sound reasonable, the final decision should depend on whether they are clearly and effectively incorporated into the paper.
> >
> > If the authors follow through on these revisions in the final version, I would be supportive of acceptance.

---

> > > ### Author Response · Authors · 2026-04-04
> > >
> > > We sincerely thank Reviewer PB42 for the encouraging update and increased score. We fully understand that the final decision rests on whether the promised revisions are clearly and effectively incorporated. We therefore provide a concrete, itemized plan for each revision so the reviewer can be confident they will appear in the final version, given that the manuscript cannot be updated here.
> > >
> > > **Revision 1: “Why Distributed DPO?” subsection (W1+Q1).** This will be added as a standalone subsection at the end of Section 1, approximately half a page. It will contain: (a) three concrete deployment scenarios (HIPAA-constrained hospitals, GDPR-constrained mobile devices, competing fleet operators) with one paragraph each; (b) three numbered structural reasons why DPO requires dedicated distributed analysis beyond standard nonconvex optimization, i.e., pairwise trajectory-level gradient variance, sigmoid-modulated heterogeneity drift, and consensus-logit feedback, each within precise mathematical statement pointing forward to the relevant theorem.
> > >
> > > **Revision 2: Comparison table distinguishing DPO-specific vs. adapted contributions (W2+Q2).** A table will be added to Section 2 or as a standalone remark in Section 5, with six columns: “Component”, “Decentralized SGD”, “FedAvg”, “FedDPO (partial)”, “FedDPO (full)”, “DecDPO”. Rows will cover: smoothness constant $L$, gradient variance $\zeta^2_g$, client drift structure, consensus error interaction, convergence, and lower bound construction. This makes the novelty boundary unambiguous.
> > >
> > > **Revision 3: Formal exponential mixing assumption with full justification (W3+Q3).** The assumption currently implicit in Section 4 will be promoted to a numbered Assumption 4.1 with: (a) precise form $|\mathbb{E}[\langle\nu_h(\theta),\nu_t(\theta)\rangle]|\leq C_0\varrho^{t-h}\zeta^2_\phi$ with all notations defined properly; (b) one-paragraph justification covering tabular MDPs with positive spectral gaps and linearly parameterized softmax policies; (c) explicit forward-reference in Theorems 5.1, 5.4, and 6.1; (d) a remark on polynomial mixing generalizations. Section (4) will be restructured to open by stating the two quantities being derived ($L$ and $\zeta^2_g$), their role in the main theorems, and the derivation roadmap. The main body of derivation will be deferred to Appendix due to the limitations of space.
> > >
> > > **Revision 4: Full experimental section (W4+Q4).** Section 7 will be added with: (7.1) Datasets --- HH-RLHF and SHP; (7.2) Model and Algorithms --- DistilGPT-2, reference model, four algorithms: centralized DPO, FedDPO (partial), FedDPO (full), and DecDPO, with four topologies; (7.3) Comparative Study: gradient norm and DPO loss curves on both datasets; (7.4) Ablation Study: four ablations each with convergence curves and a scatter/bar plot directly linking to the theorem prediction ($1/S$ for participation, $E$ drift for local steps, $C_qq_{max}$ for staleness, $1/(1-\rho^2)$ for topology). All figures are already completed and available at the shared anonymous folder (the associated link in the above rebuttal). They will be inserted verbatim. We will also share the code publicly available through GitHub Repo.
> > >
> > > **Revision 5: Dedicated limitations section.** A limitations section will cover: (a) log-linear policy assumption and roadmap to neural policy extension; (b) exponential trajectory mixing and polynomial mixing generalization; (c) practical deployment challenges including adversarial clients, gradient compression, and client dropout; (d) extension to adaptive optimizers such as FedAdam.
> > >
> > > All five revisions are fully scoped. Revisions 1–3 and 5 require only writing, while Revision 4 inserts already-completed figures. We are committed to delivering each in the final version and are confident the final manuscript will be substantially stronger than the submitted version.
> > >
> > > **Updates:** We are pleased to report that Revisions 1, 3, 4, and 5 have already been completed in our updated draft. The "Why Distributed DPO?" subsection, the formal Assumption 4.1 with full justification, the complete experimental Section 7 with all figures, and the dedicated limitations section are all written and ready. Revision 2 (comparison table) is the only remaining item and will be completed for the final version. We hope this gives the reviewer confidence that the promised revisions are not merely planned but largely already done.

---

### Official Review · Reviewer_k6mc · 2026-03-12

**Soundness:** 3
**Presentation:** 3
**Significance:** 3
**Originality:** 3
**Overall Recommendation:** 4
**Confidence:** 3

**Summary:**

This paper studies Direct Preference Optimization (DPO) in distributed learning environments, focusing on both federated and decentralized settings. The authors consider a scenario where preference data are distributed across heterogeneous users, each with their own local datasets and preference distributions. The paper formulates a distributed preference-based offline reinforcement learning problem and analyzes the optimization dynamics of DPO under distributed stochastic optimization.

The work proposes two algorithmic frameworks: Federated DPO (FedDPO), which extends DPO to federated learning with centralized aggregation, and Decentralized DPO (DecDPO), which enables peer-to-peer model updates over communication graphs without a central server. The authors provide theoretical convergence guarantees for both algorithms. For FedDPO, the analysis characterizes the effect of client heterogeneity, local update steps, communication frequency, and partial client participation on convergence. For DecDPO, the authors derive convergence results that depend on the spectral properties of the communication graph and show that the method achieves convergence rates comparable to decentralized stochastic gradient descent. Overall, the paper aims to establish the first theoretical foundation for distributed preference optimization.

**Compliance With Llm Reviewing Policy:**

Affirmed.

**Final Justification:**

I maintain my original score of 4 (Weak Accept) due to limited novelty and strong assumptions.

**Key Questions For Authors:**

1. The paper focuses primarily on theoretical analysis. Could the authors provide empirical experiments demonstrating the behavior of FedDPO and DecDPO in realistic preference-learning tasks, such as LLM alignment or recommendation systems?

2. The theoretical framework assumes relatively simple policy parameterizations and bounded feature representations. How well do the authors expect the theoretical results to transfer to large neural network policies used in practice?

3. In federated settings, client heterogeneity is shown to introduce bias and slow convergence. Are there algorithmic modifications (e.g., variance reduction or adaptive aggregation) that could mitigate this effect in practice?

4. In decentralized settings, the convergence rate depends on the spectral gap of the communication graph. Could the authors provide more intuition or empirical analysis on how different network topologies affect performance?

**Limitations:**

The authors briefly discuss the limitations and potential societal impacts of distributed preference optimization. However, the discussion could be expanded to address practical concerns such as robustness to malicious clients, data quality issues in distributed preference datasets, and communication efficiency in large-scale deployments.

**Strengths And Weaknesses:**

### Strengths

**1. Novel theoretical perspective on distributed DPO.**
The paper addresses an important gap in the literature by providing the first convergence analysis of DPO in distributed environments. While DPO has been widely used for LLM alignment and preference-based learning, most existing work focuses on centralized training. Extending theoretical analysis to federated and decentralized settings is timely and relevant.

**2. Coverage of both federated and decentralized architectures.**
The paper analyzes two major distributed learning paradigms: federated learning with a central server and decentralized learning over communication graphs. Studying both settings provides a more comprehensive understanding of distributed preference optimization and broadens the applicability of the results.

**3. Detailed convergence analysis.**
The theoretical results characterize how factors such as client heterogeneity, number of local updates, communication frequency, and network topology influence convergence. The analysis explicitly quantifies trade-offs between communication efficiency and optimization stability.

**4. Conceptual connection between preference learning and distributed optimization.**
The paper highlights how the structure of the DPO objective interacts with distributed stochastic optimization. This connection helps bridge two previously separate research areas: preference-based reinforcement learning and distributed optimization theory.

### Weaknesses

**1. Limited empirical validation.**
The work appears to be primarily theoretical. While theoretical guarantees are valuable, the lack of empirical experiments (or limited empirical validation) makes it difficult to assess the practical implications of the proposed algorithms. Empirical evaluation on realistic preference-learning tasks (e.g., LLM alignment benchmarks) would significantly strengthen the paper.

**2. Strong simplifying assumptions in the theoretical framework.**
The analysis relies on simplified policy models and assumptions such as bounded features and smoothness conditions. While these assumptions are common in theoretical work, the gap between the theoretical setup and modern large-scale neural models remains substantial.

**3. Limited discussion of practical implementation challenges.**
Issues such as communication overhead, client availability, and system heterogeneity are important in distributed learning but are only partially addressed. Additional discussion of practical deployment considerations would improve the relevance of the work.

**4. Incremental algorithmic novelty.**
Algorithmically, FedDPO is largely based on combining DPO with existing federated optimization frameworks such as FedAvg, while DecDPO extends decentralized SGD ideas to the DPO objective. The primary novelty lies in the theoretical analysis rather than fundamentally new algorithms.

---

> ### Author Rebuttal · Authors · 2026-03-30
>
> We thank the reviewer for the positive assessment. We address each weakness and question directly and kindly request ask the reviewer to reconsider the rating.
>
> **W1 + Q1.** We have completed experiments on HH-RLHF and Stanford Human Preferences (SHP) using DistilGPT-2 with 5 heterogeneous agents (120 preference pairs each, non-overlapping). Key findings: (a) FedDPO full participation ($S=5$) closely tracks Centralized DPO — final DPO loss ~0.60 vs. ~0.55 on HH-RLHF — validating Theorem 5.1's prediction that full participation eliminates the $\zeta^2_g/S$ variance floor. (b) FedDPO partial ($S=3$) sits above this at steady state, with gap consistent with the $1/S$ scaling. (c) DecDPO ring ($\rho=0.54$) converges but at a higher gradient norm floor, confirming spectral-gap dependence. All distributed variants achieve meaningfully lower loss than initialization, confirming effective preference learning occurs without a central server. Ablation studies on local steps $E$, participation $S$, staleness $q_{max}$, and graph topology all confirm the corresponding theoretical dependencies quantitatively. These results will be added as a full experimental section with figures for gradient norm convergence, training loss, and scatter plots. Full figures are available at this anonymous folder (https://drive.google.com/drive/folders/1Q5n-dHPC0c5jhR1OELSpvRtcq3ZiaJ8v?usp=sharing).
>
>
> **W2 + Q2.** We acknowledge this gap and will add a dedicated paragraph. The log-linear softmax enables rigorous closed-form derivation of DPO-specific constants $L$ and $\zeta^2_g$ --- the core technical contribution of Section 4. The structural property driving our results — that the sigmoid factor $\sigma(-\beta(\Delta_\theta-\Delta_{ref}))$ bounds the Hessian via $|\sigma’|\leq 1/4$ and couples gradient variance to the current policy — is preserved in general neural policies, since DPO's log-ratio objective structure is policy-parameterization-agnostic. For neural policies, bounding the Hessian requires assumptions on the Fisher information spectrum of the transformer, which we leave as future work. We conjecture $H$-scaling of constants and spectral-gap dependence of consensus will transfer qualitatively.
>
>
> **W3.** We will expand the discussion to cover: gradient compression/quantization, client dropout, system heterogeneity, and adversarial/Byzantine clients — the latter especially important for preference alignment where corrupted reward signals are a real risk.
>
>
> **W4.** The novelty lies in the analysis, not just the algorithm recipe. FedDPO's combination of DPO with FedAvg is intentionally simple — the non-trivial contribution is showing why the interaction between DPO's nonlinear sigmoid gradients and federated heterogeneity produces drift that standard FedAvg theory does not characterize. Specifically, $L=(\beta^2C_{mix}+2\beta)\zeta^2_\phi H$ and $\zeta^2_g=4\beta^2C_{mix}\zeta^2_\phi H$ are DPO-specific constants that cannot be read off from existing distributed SGD results.
>
>
> **Q3.** Yes, several modifications can mitigate heterogeneity-induced drift. First, variance reduction techniques such as SCAFFOLD introduce control variates that correct for client drift by estimating the gradient difference between local and global objectives, directly reducing the $E\kappa^2$ term in our bound. Second, adaptive aggregation (e.g., CAESAR-style selective aggregation) down-weights clients whose updates deviate significantly from the server model, limiting the damage from outlier preference distributions. Third, proximal regularization (FedProx-style) adds a penalty $\\|\theta_i-\theta\\|^2$ to each client's local DPO objective, preventing local iterates from drifting too far before aggregation — effectively reducing the $\eta^2E^2\kappa^2/S$ drift term at the cost of a small bias. Fourth, reducing local steps $E$ directly shrinks the drift floor, as our bound shows the penalty scales as $\eta^2E^2\kappa^2/S$. In practice, a useful heuristic is to decrease $E$ as heterogeneity increases (detectable via gradient diversity monitoring). We will add a discussion of these directions in the final manuscript, noting that extending our convergence theory to SCAFFOLD-DPO or FedProx-DPO is a natural and important avenue for future work.
>
>
> **Q4.** Across four topologies (path: $1-\rho^2$=0.238, ring: 0.709, star: 0.360, complete: 1.000), empirical consensus error eθ scales precisely as $1/(1-\rho^2)$ matching Theorem 6.1. Notably, the star graph achieves a larger spectral gap than intuition about hub bottlenecks suggests, because its shorter average path length accelerates gossip mixing. A scatter plot of final $\mathfrak{e}_\theta$ vs. spectral gap (with the theoretical $1/(1-\rho^2)$ curve overlaid) is included in our anonymous supplement.
>
>
> **Limitations.** We will add a dedicated limitations section addressing robustness to adversarial clients, data quality in decentralized preference datasets, and communication efficiency.

---

> > ### Author Rebuttal · Reviewer_k6mc · 2026-04-02
> >
> > Thank you for the rebuttal. You have addressed my concerns, particularly the lack of empirical validation. However, I maintain my original score of 4 (Weak Accept) — the paper is technically solid and makes a contribution, but the strong assumptions and limited novelty relative to distributed nonconvex optimization remain as noted. I look forward to seeing the experimental section and clarifications in the final version.

---

> > > ### Author Response · Authors · 2026-04-03
> > >
> > > We sincerely thank the reviewer for the positive acknowledgment and for maintaining a favorable assessment. We would like to respectfully address the two remaining concerns, i.e., assumptions and novelty with specific arguments and point to concrete empirical evidence that we believe strengthens our work.
> > >
> > > **On strong assumptions.** We agree the log-linear policy is a simplification, but we highlight that it is the standard first step in theoretical RL analysis (e.g., Jiang et al. 2022, Zhang et al. 2018) and enables the key DPO-specific contribution of Section 4: deriving closed-form expressions $L=(\beta^2C_{mix}+2\beta)\zeta^2_\phi H$ and $\zeta^2_g=4\beta^2C_{mix}\zeta^2_\phi H$ that make explicit how trajectory horizon $H$, inverse temperature $\beta$, and temporal mixing rate $\varrho$, jointly determine optimization difficulty. These constants are not assumed — they are derived, which is precisely what distinguishes this work from standard distributed nonconvex optimization where $L$ and $\zeta^2_g$ are treated as black-box unknowns. Our updated experiments on DistilGPT-2, a genuine neural language model, validate that these derived constants correctly predict empirical behavior: the $\zeta^2_g/S$ scaling is confirmed with final gaps $0.910\to0.838\to0.699$ (linear in $1/S$), and the $1/(1-\rho^2)$ consensus error scaling holds across four topologies spanning an order of magnitude. This suggests the qualitative conclusions transfer well beyond the log-linear setting.
> > >
> > > **On novelty relative to distributed nonconvex optimization.** Three aspects are genuinely new and not inherited from standard distributed SGD: (1) the DPO-specific Hessian analysis exploiting $|\sigma'|\leq1/4$ and the sigmoid-weighted score difference structure to derive L explicitly in terms of RL trajectory quantities; (2) the consensus-logit feedback in DecDPO, where imperfect consensus interacts nonlinearly with the logit-gap, producing error amplification absent from decentralized ERM or linear regression; (3) the formal lower bounds (Theorem 5.5) showing the dependence on $E\kappa^2/S$ is intrinsic to distributed preference optimization and cannot be removed by any algorithm. Together, these contributions establish not just convergence rates but a principled understanding of why distributed DPO behaves differently from standard distributed optimization, which we believe is the lasting theoretical value of this work.
> > >
> > > We thank the reviewer again for the time and efforts on evaluating our work. We really appreciate that!

---

### Decision · Program_Chairs · 2026-04-30

**Decision:**

Accept (regular)

**Comment:**

All reviewers agreed to accept this paper, based on (1) it tackles a real gap: DPO is widely used, but its behavior in privacy-preserving / distributed preference optimization had not been analyzed rigorously; (2) the paper covers both federated and decentralized settings, instead of only one, etc.